# OPTIMIZATION ON MULTIPLE MANIFOLDS

## ABSTRACT

Optimization on manifold has been widely used in machine learning, to handle optimization problems with constraint. Most previous works focus on the case with a single manifold. However, in practice it is quite common that the optimization problem involves more than one constraints, (each constraint corresponding to one manifold). It is not clear in general how to optimize on multiple manifolds effectively and provably especially when the intersection of multiple manifolds is not a manifold or cannot be easily calculated. We propose a unified algorithm framework to handle the optimization on multiple manifolds. Specifically, we integrate information from multiple manifolds and move along an ensemble direction by viewing the information from each manifold as a drift and adding them together. We prove the convergence properties of the proposed algorithms. We also apply the algorithms into training neural network with batch normalization layers and achieve preferable empirical results.

## 1 INTRODUCTION

Machine learning problem is often formulated as optimization problem. It is common that the optimization problem comes with multiple constraints due to practical scenarios or human prior knowledge that adding some of them help model achieve a better result. One way to handle these constraints is adding regularization terms to the objective, such as the $\ell_1$ and $\ell_2$ regularization. However, it is hard to adjust the hyper-parameters of the regularization terms to guarantee that the original constraints get satisfied.

Another way to deal with the constraints is to optimize on manifolds determined by the constraints. Then the optimization problem becomes unconstrained on the manifold, which could be easy to solve technically. Furthermore, optimization on manifold indicates optimizing on a more compact space, and may bring performance gain when training neural networks, e.g., (Meng et al., 2018; Cho & Lee, 2017).

Most previous works on manifold optimization focus on a single manifold (Zhang & Sra, 2016; Wei et al., 2016). However, in practice, we often face more than one constraints, each of them corresponding to one manifold. If we still solve the optimization problem with multiple constraints by method on manifold, we need to handle it on the intersection of multiple manifolds, which may no longer be a manifold (Pathak et al., 2015). Due to this, traditional optimization methods on manifold does not work in this case.

In this paper, we consider the problem of optimization on multiple manifolds. Specifically, the problem is written as

$$\underset{x \in \bigcap_{i=1}^{n} \mathcal{M}_i}{\arg\min} \; f(x), \tag{1}$$

where each $\mathcal{M}_i$ is a manifold. We propose a method solving this problem by choosing the moving direction as $-\nabla f(x)$ (on manifold is $-\mathrm{grad} f(x)$) with several drifts which are derived from the descent information on other manifolds. By this method, we get sequence that has information from all manifolds.

### 1.1 RELATED WORK

There are several articles discussing the problem of optimization on manifold. Most of them focus on a single manifold. Readers can find a good summary about this topic and the advantages of op-

timization on manifold in (Absil et al., 2009). Recently, popular first order algorithms in Euclidean space are studied in the manifold setting, e.g., the convergence of gradient descent (Zhang & Sra, 2016; Boumal et al., 2017), sub-gradient method (Hosseini & Uschmajew, 2017), stochastic variance reduction gradient (SVRG) (Zhang et al., 2016) and the gradient descent with momentum (Liu et al., 2017).

Riemann approaches (Cho & Lee, 2017; Huang et al., 2017) have also been applied to train deep neural network by noticing that the parameters of the neural network with batch normalization live on Grassmann manifold and Oblique manifold, respectively.

## 1.2 CONTRIBUTION

($\mathbf{1}$) This paper introduces an algorithm to deal with optimization with multiple manifolds. The algorithm adds drifts obtained from other manifolds to the moving direction, in order to incorporate the information from multiple manifolds during the optimization process.

($\mathbf{2}$) We prove the convergence of this algorithm under a very general framework. The proof is also applicable to the convergence of many other algorithms including gradient descent with momentum and gradient descent with regularization. Moreover, our proof does not depend on the choices of $\text{Retr}_x$ on the manifold.

## 2 OPTIMIZATION WITH MULTIPLE MANIFOLDS

### 2.1 BASIC DEFINITION AND LEMMA

The specific definition of manifold $\mathcal{M}$ can be found in any topology book. For better understanding, we introduce several properties of manifold here. A manifold is a subspace of $\mathbb{R}^n$. For a given point $x \in \mathcal{M}$, it has a tangent space $T_x \mathcal{M}$ which is a linear space but $\mathcal{M}$ may not. For gradient descent method, the iterates are generated via

$$x_{k+1} = x_k - \eta \nabla f(x_k),$$

$\eta$ is step length. However, the iterate generated by gradient descent may not on manifold anymore because manifold is not a linear space. To fix this, we introduce a retraction function $\text{Retr}_x(\eta)$ : $T_x \mathcal{M} \to \mathcal{M}$ to determine how point moves on manifold. Specifically, if $\mathcal{M}$ is $\mathbb{R}^n$, the $\text{Retr}_x$ becomes $x + \eta$. We can consider $\eta$ in $\text{Retr}_x$ as the moving direction of the iterating point. Then, the gradient descent on manifold (Boumal et al., 2017; Zhang & Sra, 2016) is given by

$$x_{k+1} = \text{Retr}_x(-\frac{1}{L}\text{grad}f(x_k)), \tag{2}$$

where $\text{grad}f(x)$ is Riemannian gradient. Riemannian gradient is the orthogonal projection of gradient $\nabla f(x)$ to tangent space $T_x \mathcal{M}$ as $\nabla f(x)$ may not in tangent space $T_x \mathcal{M}$ and the moving direction on manifold is only decided by the vector in $T_x \mathcal{M}$. All of notations related to manifold can be referred to (Absil et al., 2009).

We next use a lemma to describe a property of the minimum point of the problem $\arg\min_{x \in \mathcal{M}} f(x)$, which is a special case of Yang et al., 2014, Corollary 4.2 and Boumal et al., 2017, Proposition 1.

**Lemma 2.1** *Let $x$ be a local optimum for the optimization problem $\arg\min_{x \in \mathcal{M}} f(x)$, which means there exists a neighborhood $U_x$ of $x$ satisfy $f(x) \le f(y)$ for $y \in U_x$. If $f(x)$ is differential in $x$, then $\|\text{grad}f(x)\| = 0$.*

We see that $\text{grad}f(x)$ plays a role of $\nabla f(x)$ on manifold. Similar as Boumal et al. (2017) discussed, we assume function has the property of Lipschtiz gradient. The definition of Lipschtiz gradient is

**Definition 2.1 (Lipschtiz gradient)** *For any two points $x, y$ in the manifold $\mathcal{M}$, $f(x)$ satisfy:*

$$f(y) \le f(x) + \langle \text{grad}f(x), \text{Retr}_x^{-1}(y) \rangle + \frac{L}{2} \|\text{Retr}_x^{-1}(y)\|^2$$

*Then we say that $f$ satisfies the Lipschtiz gradient condition.*

We next introduce a condition that guarantees the convergence of iterative algorithms.

**Definition 2.2 (Descent condition)** *For a sequence $\{x_k\}$ and $a_k > 0$, if*

$$f(x_k) - f(x_{k+1}) \geq a_k \|\mathrm{grad} f(x_k)\|^2,$$

*then we say the sequence satisfies the descent condition.*

## 2.2 GRADIENT DESCENT ON TWO MANIFOLDS

First, we introduce a theorem to describe the convergence when the object function $f$ is lower finite, i.e., there exists a $f^*$ such that $f(x) \geq f^* > -\infty$ for all $x$, and the iterates satisfy descent condition. This theorem plays a key role in proof of the rest theorems.

**Theorem 2.1** *If $f$ is lower finite, and the iteration sequence $\{x_k\}$ satisfies the descent condition for any given $\{a_k\}$, where each $a_k > 0$. Then $\liminf_{k\to\infty} a_k \|\mathrm{grad} f(x_k)\| = 0$*

**Proof 1** *The proof is available in Supplemental.*

For better presentation, we first describe the algorithm under the circumstance of two manifolds. Considering the objective function $f$ constrained on two manifolds $\mathcal{M}_1, \mathcal{M}_2$, we aim to find the minimum point on $\mathcal{M}_1 \bigcap \mathcal{M}_2$. Since $\mathcal{M}_1 \bigcap \mathcal{M}_2$ may not be a manifold, previous methods on manifold optimization cannot apply directly. We propose a method that integrates information from two manifolds over the optimization process.

Specifically, we construct two sequences $\{x_k\}, \{y_k\}$, each on one manifold respectively. We add a drift which contains information from the other manifold to the original gradient descent on manifold (equation 2). The updating rules are

$$x_{k+1} = \mathrm{Retr}_{x_k}[-\frac{1}{L}(a_k^{(1)}\mathrm{grad} f(x_k) + b_k^{(1)} h_k^{(1)})], \tag{3}$$

$$y_{k+1} = \mathrm{Retr}_{y_k}[-\frac{1}{L}(a_k^{(2)}\mathrm{grad} f(y_k) + b_k^{(2)} h_k^{(2)})] \tag{4}$$

If $b_k = 0$ in (equation 3) and (equation 4), the updating rules reduce to normal gradient descent on manifold equation 2. The drift $h_k$ is in the tangent space $T_x\mathcal{M}$ of each manifold, which represents information from the other manifold. We call this algorithm *gradient descent on manifold with drifting*, whose procedure is described in Algorithm 1.

---

**Algorithm 1** Gradient descent with drift on manifold

Input $\delta > 0$, $x_0 \in \mathcal{M}_1$, $y_0 \in \mathcal{M}_2$, $\mathrm{Retr}_x$, $\varepsilon > 0$
$k \to 0$
**while** $\|\mathrm{grad} f(x_k)\| > \varepsilon$ or $\|\mathrm{grad} f(y_k)\| > \varepsilon$ **do**
  $\delta \leq a_k^{(1)} \leq 2$
  $\delta \leq a_k^{(2)} \leq 2$
  Calculating $\mathrm{grad} f(x_k) = P_{x_k}^{(1)}\nabla f(x_k), \mathrm{grad} f(y_k) = P_{y_k}^{(2)}\nabla f(y_k)$.
  $P_{x_k}^{(1)}$ and $P_{y_k}^{(2)}$ are respectively projection matrix of tangent space $T_{x_k}, T_{y_k}$.
  Obtain $h_k^{(1)} \in T_{x_k}\mathcal{M}_1$ and $h_k^{(2)} \in T_{y_k}\mathcal{M}_2$.
  $b_k^{(1)} = \frac{2(1-a_k^{(1)})\langle \mathrm{grad} f(x_k), h_k^{(1)}\rangle}{\|h_k^{(1)}\|^2}$
  $b_k^{(2)} = \frac{2(1-a_k^{(2)})\langle \mathrm{grad} f(y_k), h_k^{(2)}\rangle}{\|h_k^{(2)}\|^2}$
  $x_{k+1} = \mathrm{Retr}_{x_k}[-\frac{1}{L}(a_k^{(1)}\mathrm{grad} f(x_k) + b_k^{(1)} h_k^{(1)})]$     update step
  $y_{k+1} = \mathrm{Retr}_{y_k}[-\frac{1}{L}(a_k^{(2)}\mathrm{grad} f(y_k) + b_k^{(2)} h_k^{(2)})]$     update step
  $k \leftarrow k + 1$
**end while**
**return** $x_k, y_k$

---

We next present the convergence theorem of this algorithm, which illustrates how we set $a_k$ and $b_k$ in the algorithm.

**Theorem 2.2** *For function $f(x)$ is lower finite, and Lipschtiz gradient. If we construct the sequence $\{x_k\}$ like equation (3), and for any $0 < \delta < 2$, we control $\delta \leq a_k \leq 2$. Setting*

$$b_k = \frac{2(1 - a_k)\langle \mathrm{grad} f(x_k), h_k \rangle}{\|h_k\|^2}, \tag{5}$$

*then $x_k$ convergence to a local minimizer.*

**Proof 2** *The proof is based on construction of the descent condition (equation 12) and is available in Supplemental.*

From the construction of $b_k$, we can see that the smaller the correlation between $\mathrm{grad} f(x_k)$ and $h_k$ is, the smaller effect the information from $\mathcal{M}_2$ brings. In fact, we set $h_k^{(1)} :=$ $\|\mathrm{grad} f(x_k)\| \frac{P_{x_k}^{(1)} \mathrm{grad} f(y_k)}{\|P_{x_k}^{(1)} \mathrm{grad} f(y_k)\|}$, where $P_{x_k}^{(1)}$ is the projection matrix to tangent space $T_{x_k} \mathcal{M}_1$. Similarly we set $h_k^{(2)}$ which exchanges $x_k$ and $P_{x_k}^{(1)}$ with $y_k$ and $P_{y_k}^{(2)}$ (projection matrix of tangent space $T_{y_k} \mathcal{M}_2$). The drift intuitively gives $x_k$ a force moving towards the minimizer on the other manifold. If the two manifolds are $\mathbb{R}^n$, then $x_k$ and $y_k$ are symmetry with each other. We have

$$\begin{cases} x_{k+1} = x_k - \frac{1}{L}(a_k^{(1)} \nabla f(x_k) + b_k^{(1)} h_k^{(1)}) \\ y_{k+1} = y_k - \frac{1}{L}(a_k^{(2)} \nabla f(y_k) + b_k^{(2)} h_k^{(2)}). \end{cases} \tag{6}$$

If the equation system is stable and $x_0, y_0$ are mutually close, the distance between $x_k$ and $y_k$ will be small when $k \to \infty$. By Schwarz inequality, we see $b_k \leq 2(1 - a_k)$. Since $\|h_k\| = \|\mathrm{grad} f(x_k)\|$, the scale of the drift is the same as the original Riemannian gradient. Hence, information from another manifold will not affect much, when the points $x_k$ and $y_k$ are close to a minimizer. We can control the contribution of the information from the other manifold by adjusting $a_k$. For instance, $a_k = 1$ indicates we do not integrate information from the other manifold.

We can also prove the convergence rate of this algorithm.

**Theorem 2.3** *If $f$ is lower finite satisfy $f(x) \geq f^* > -\infty$, we choose $a_k$ as $\max\{\frac{1}{k+1}, \delta\}, 0 < \delta < 2$. Then, for any $\varepsilon > 0$, making sure $\|\mathrm{grad} f(x_k)\| < \varepsilon$ needs at most $(\lceil \exp\{\frac{f(x_0) - f^*}{\varepsilon^2} + \frac{\pi^2}{12}\} \rceil - 1) \wedge \lceil \frac{2(f(x_0) - f^*)}{\varepsilon^2(2\delta - \delta^2)} \rceil$ iterations.*

**Proof 3** *The proof is delegated to Supplemental.*

Theorem 2.3 states the number of iterations we need to achieve a specific accuracy. Here we can adjust $a_k$ as long as $\delta < a_k < 2$.

### 2.3 GRADIENT DESCENT ON $n$ MANIFOLDS

In this subsection, we describe our algorithm for the case with multiple (more than 2) manifolds. Suppose we have $n$ manifolds, $\mathcal{M}_1, \cdots, \mathcal{M}_n$, and sequence on manifold $\mathcal{M}_i$ is denoted as $\{x_k^{(i)}\}$. In the following, we use sequence $\{x_k^{(1)}\}$ on $\mathcal{M}_1$ as an example, and other sequences on other manifolds can be derived accordingly. Let $g_k^{(i)} \in T_x \mathcal{M}_1$ denote the drift from manifold $\mathcal{M}_i$ ($g_k^{(1)}$ is $\mathrm{grad} f(x_k^{(1)})$). Then let the updating rule be

$$x_{k+1}^{(1)} = \mathrm{Retr}_{x_k}\left[ -\frac{1}{L} \sum_{i=1}^{n} a_k^{(i)} g_k^{(i)} \right]. \tag{7}$$

Since $f$ satisfies Lipschtiz gradient condition(2.1), we have

$$f(x_k^{(1)}) - f(x_{k+1}^{(1)}) \geq \frac{1}{L} \left\langle g_k^{(1)}, \sum_{i=1}^n a_k^{(i)} g_k^{(i)} \right\rangle - \frac{1}{2L} \left\| \sum_{i=1}^n a_k^{(i)} g_k^{(i)} \right\|^2$$

$$= \frac{1}{L} \{ (a_k^{(1)} - \frac{a_k^{(1)2}}{2}) \langle g_k^{(1)}, g_k^{(1)} \rangle + \sum_{i=2}^n [a_k^{(i)} \langle g_k^{(1)}, g_k^{(i)} \rangle - \frac{1}{2} a_k^{(1)} a_k^{(i)} \langle g_k^{(1)}, g_k^{(i)} \rangle$$

$$- \frac{1}{2} \sum_{j=1}^n a_k^{(i)} a_k^{(j)} \langle g_k^{(i)}, g_k^{(j)} \rangle ] \}.$$

We choose $a_k^{(1)}$ such that $0 < \delta < a_k^{(1)} < 2$. For $j = 2 \cdots n$, we choose $a_k^{(j)}$ such that

$$\sum_{j=2}^n a_k^{(j)} \langle g_k^{(i)}, g_k^{(j)} \rangle = 2(1 - a_k^{(1)}) \langle g_k^{(1)}, g_k^{(i)} \rangle \qquad \text{for } i = 2 \cdots n \tag{8}$$

then

$$a_k^{(i)} \langle g_k^{(1)}, g_k^{(i)} \rangle - \frac{1}{2} \sum_{j=1}^n a_k^{(i)} a_k^{(j)} \langle g_k^{(i)}, g_k^{(j)} \rangle - \frac{1}{2} a_k^{(1)} a_k^{(i)} \langle g_k^{(1)}, g_k^{(i)} \rangle = 0 \qquad \text{for } i = 2 \cdots n.$$

The way of choosing $a_k^{(j)}$ is to obtain the descent condition. Specifically, when $n = 2$, the solution of (equation 8) becomes $b_k$ in (equation 5). If $a_k^{(j)}$ solve the linear equation system (8), we get the descent condition (equation 12). From theorem 2.1, we prove the convergence of the updating rule (equation 7) for the case with $n$ manifolds. Writing (equation 8) in the matrix form, we have

$$G_k \alpha_k = \beta_k,$$

where $G_k = (\langle g_k^{(i)}, g_k^{(j)} \rangle)_{ij}$, $i, j$ from 2 to $n$, and $\alpha_k = (a_k^{(2)}, \cdots, a_k^{(n)})^T, \beta_k = (2(1 - a_k^{(1)}) \langle g_k^{(1)}, g_k^{(2)} \rangle, \cdots, 2(1 - a_k^{(1)}) \langle g_k^{(1)}, g_k^{(n)} \rangle)^T$. If $G_k$ is invertible, the linear equation system has an unique solution.

# 3 APPLY OUR ALGORITHM TO TRAIN NEURAL NETWORK

## 3.1 NEURAL NETWORK WITH BATCH NORMALIZATION

Batch normalization has been widely used since its proposition (Ioffe & Szegedy, 2015). It transforms the input value to a neuron from $z = w^T x$ to

$$BN(w) = \frac{z - E(z)}{\sqrt{Var(z)}} = \frac{w^T(x - E(x))}{\sqrt{w^T V_x w}}.$$

We can calculate the derivative as follows

$$\frac{\partial BN(w)}{\partial w} = \frac{x - E(x)}{\sqrt{w^T V_x w}} - \frac{w^T(x - E(x)) V_x w}{(w^T V_x w)^{\frac{3}{2}}}.$$

For any $a \neq 0, a \in \mathbb{R}$, we see that $BN(w) = BN(aw)$ and $\frac{\partial BN(aw)}{\partial aw} = \frac{1}{a} \frac{\partial BN(w)}{\partial w}$. These equations mean that after a batch normalization, the scale of parameter has no relationship with the output value, but scale of gradient is opposite with the scale of parameter. Cho & Lee (2017) have discussed that batch normalization could have an adverse effect in terms of optimization since there can be an infinite number of networks, with the same forward path but different scaling, which may converge to different local optima owing to different gradients.

To avoid this phenomenon, we can eliminate the effect of scale by considering the weight $w$ on the Grassmann manifold or Oblique manifold. On these two manifolds, we can ignore the scale of parameter. Cho & Lee (2017); Huang et al. (2017) respectively discuss that $BN(w)$ has same image space on $\mathcal{G}(1, n)$ and $St(n, 1)$ as well as $\mathbb{R}^n$, where $\mathcal{G}(1, n)$ is a Grassmann manifold and $St(n, 1)$ is an Oblique manifold. Due to these, we can consider applying optimization on manifold to batch

normalization problem. However, the property of these two manifold implies that we can actually optimize on the intersection of two manifolds. Since optimization on a manifold rely on Riemannian gradient $\mathrm{grad} f(x)$ and $\mathrm{Retr}_x$, for a specific $\mathrm{Retr}_x$ (9) of Grassmann manifold $\mathcal{G}(1, n)$, we get a unit point $x$ when $\eta = -\mathrm{grad} f(x) = 0$ in formula (9). The condition $\|\mathrm{grad} f(x)\| = 0$ means we obtain a unit critical point on Grassmann manifold which is also on Oblique manifold.

The specific discussion of Grassmann manifold and Oblique manifold can be found in (Absil et al., 2009). $\mathcal{G}(1, n)$ is a quotient manifold defined on a vector space, it regards vector with same direction as same element. For example $(1, 1, 1)$ and $(10, 10, 10)$ correspond to same element. We represent elements on $\mathcal{G}(1, n)$ with same direction by choosing one of them as representation element. Oblique manifold is given by $\mathrm{St}(n, p) = \{X \in \mathbb{R}^{n \times p} : \mathrm{ddiag}(X^T X) = I_p\}$, where $\mathrm{ddiag}(\cdot)$ is diagonal matrix of a matrix.

We have discussed above that iteration point on $\mathcal{G}(1, n)$ would be a unit point when it's a local minimizer. Due to this, the local minimizer we find is actually live on the intersection of $\mathrm{St}(n, 1)$ and $\mathcal{G}(1, n)$. Hence, training neural network with batch normalized weights can be converted to the problem

$$\underset{x \in \mathcal{G}(1,n) \bigcap \mathrm{St}(n,1)}{\arg \min} f(x).$$

Let Riemannian gradient be projection of $\nabla f(x)$ to tangent space of $x$. On $\mathcal{G}(1, n)$, we have

$$P_x^{(1)}(\eta) = \eta - x^T \eta \frac{x}{\|x\|^2}$$

$$\mathrm{grad} f(x) = P_x^{(1)}(\nabla f(x)) = \nabla f(x) - (x^T \nabla f(x)) \frac{x}{\|x\|^2}$$

$$\mathrm{Retr}_x^{(1)}(\eta) = \frac{x}{\|x\|} \cos \|\eta\| + \frac{\eta}{\|\eta\|} \sin \|\eta\| \tag{9}$$

On $\mathrm{St}(n, 1)$, we have

$$P_x^{(2)}(\eta) = \eta - x \mathrm{ddiag}(x^T \eta)$$

$$\mathrm{grad} f(x) = P_x^{(2)}(\nabla f(x)) = \nabla f(x) - x \mathrm{ddiag}(x^T \nabla f(x))$$

$$\mathrm{Retr}_x^{(2)}(\eta) = \frac{x + \eta}{\|x + \eta\|}$$

the $P_x$ is the projection matrix onto the tangent space at $x$. These results can be derived from the general formulas from (Absil & Gallivan, 2006) and (Edelman et al., 1999).

In backward process of training neural network, weight parameter of each layer is a matrix. Hence, we get gradient to a matrix in every layer. To make calculation easier, we treat the gradient matrix and parameters matrix as vector. For example a $m \times n$ gradient matrix can be viewed as a $m \times n$ dimensional vector. Then we apply Algorithm 1 to update parameters, which means we optimize on a product manifold

$$\mathcal{G}(1, k_1) \times \cdots \mathcal{G}(1, k_n), \qquad \mathrm{St}(k_1, 1) \times \cdots \mathrm{St}(k_n, 1)$$

$k_i$ is number of parameters for the $i$-th hidden layer, and $n$ is number of hidden layers. We need to operate algorithm for parameter vector on each hidden layer. In other words, we update parameters layer by layer.

## 3.2 EXPERIMENT

In this section, we use data set CIFAR-10 and CIFAR-100 (Krizhevsky & Hinton, 2009) to test our algorithm. These two data sets are color images respectively have 10 and 100 classes, each of them has 50,000 training images and 10,000 test images. The deep neural network we used is Wide-ResNet (Zagoruyko & Komodakis, 2016), it output a vector which describe the probability of a data divided into each class.

In every hidden layer of neural network, we apply batch normalization to weight parameters and treat them as a vector. We have already discussed that minimizers of a neural network with batch normalized weights live on the intersection of Grassmann manifolds and Oblique manifold. Hence, we can train neural network with batch normalized weights by our algorithm(1). The biases of every

hidden layer is unrelated to batch normalization and are updated by SGD. For every training step, we calculate mean loss $\frac{1}{S}\sum_{x_i \in S} l(f(x_i, \theta), y_i)$ of a mini batch to substitute the real loss function $\mathbb{E}_x[l(f(x, \theta), y)]$, where $S$ is batch size.

The process of algorithm on two manifolds follows Algorithm 1, where the two manifolds are $\mathcal{G}(1, n)$ and $\mathrm{St}(n, 1)$, respectively. In Algorithm 1, we choose

$$h_k^{(1)} = \|\mathrm{grad} f(x_k)\| \frac{P_{x_k}^{(1)} \mathrm{grad} f(y_k))}{\|P_{x_k}^{(1)} \mathrm{grad} f(y_k)\|}, \quad h_k^{(2)} = \|\mathrm{grad} f(y_k)\| \frac{P_{y_k}^{(2)} \mathrm{grad} f(x_k))}{\|P_{y_k}^{(2)} \mathrm{grad} f(x_k)\|}$$

and $a_k^{(1)} = a_k^{(2)} = \max\{\frac{1}{k+1}, \delta\}$. In the updating rules of $x_k$ and $y_k$, we add a norm-clip to vectors $(a_k^{(1)} \mathrm{grad} f(x_k) + b_k^{(1)} h_k^{(1)})$ and $(a_k^{(2)} \mathrm{grad} f(y_k) + b_k^{(2)} h_k^{(2)})$. Then we times $\eta$ to the two vectors, where $\eta$ is the learning rate.

In the experiments, we compare three methods: 1) stochastic gradient descent on manifold with drifting (Drift-SGDM), 2) stochastic gradient descent on manifold Boumal et al. (2017) (SGDM), and 3) stochastic gradient descent (SGD). In Algorithm 1, we can get two sequences each corresponding to a model on a manifold. We predict output class by adding two output vectors of two models and choosing the biggest as prediction class.

For Drift-SGDM (Algorithm 1), we set $\delta = 0.9$ and initial learning rate $\eta_m = 0.4$ for weights parameters which is multiplied by 0.4 at 60, 120, and 160 epochs. Initial learning rate $\eta$ for biases is 0.01 which is multiplied by 0.4 at 60, 120, and 160 epochs. Norm clip is 0.1. Training batch size is 128. The number of training epochs is 200.

For SGDM, we choose $a = 1$ in Algorithm 1. The other settings are the same as Drift-SGDM. That $a = 1$ in Algorithm 1 means that SGDM optimizes on each manifold individually

We set SGD as baseline. The learning rate is 0.2 which is multiplied by 0.2 at epoch 60,120 and 160. Weight decay is set as 0.0005, but we do not apply weight decay for algorithms on manifold. All other settings are the same as the above two algorithms.

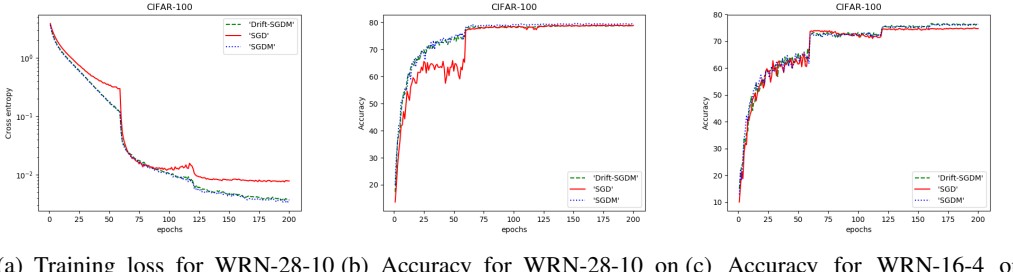

(a) Training loss for WRN-28-10 on CIFAR-100    (b) Accuracy for WRN-28-10 on CIFAR-100    (c) Accuracy for WRN-16-4 on CIFAR-100

Figure 1: Some results based on Wide-ResNet. WRN-$d$-$k$ denotes a wide residual network that has $d$ convolutional layers and a widening factor $k$

About Drift-SGDM and SGDM, the loss is achieved from the average of two model. The parameter scale of the two model can be different, because they respectively live on Grassmann manifold and Oblique manifold. Due to this, the comparing between Drift-SGDM and SGDM is more reasonable. We also give the accuracy curve and a tubular of accuracy rate on test sets to validate our algorithms.

| Dataset | CIFAR-10 | | | CIFAR-100 | | |
|---|---|---|---|---|---|---|
| Model | SGD | Drift-SGDM | SGDM | SGD | Drift-SGDM | SGDM |
| WRN-52-1 | 93.3 | 92.88 | 93.11 | 71.07 | 69.61 | 69.58 |
| WRN-16-4 | 94.46 | 94.16 | 94.16 | 74.74 | 76.4 | 76.04 |
| WRN-28-10 | 95.59 | 95.07 | 94.79 | 78.88 | 78.92 | 79.37 |

Table 1: Accuracy rate on test sets of data.

We see that our algorithm perform better on larger neural network. Our algorithm does not have regularization term, and it does not perform well in the aspect of generalization. We can actually add a regularization term like in (Cho & Lee, 2017) to achieve better generalization.

We choose $\delta$ in Algorithm 1 as 0.9. Since $b_k^{(i)} \leq 2(1 - a_k^{(i)})$ where $i = 1, 2$ as we have discussed in section 2, we see drift term $b_k^{(i)} h_k^{(i)}$ in Algorithm 1 doesn't affect much to iteration point. We can actually set a smaller $\delta$ to enhance the influence of drift term $b_k^{(i)} h_k^{(i)}$.

## 4 CONCLUSION

In this paper, we derive an intuitively method to approach optimization problem with multiple constraints which corresponds to optimizing on the intersection of multiple manifolds. Specifically, the method is integrating information among all manifolds to determine minimum points on each manifold. We don't add extra conditions to constraints of optimization problem, as long as each constraint can be converted to a manifold. In the future, we may add some conditions to manifolds which derive a conclusion that minimum points on each manifold achieved by our algorithm are close with other. If this conclusion is established, the problem of optimization on intersection of multiple manifolds is solved.

According to the updating rule (equation 3), we can derive many other algorithms, because the drift $h_k$ in (equation 3) is flexible. On the other hand, $\mathrm{Retr}_x$ on our algorithm does not limit to a specific one. Since there are some results for $\mathrm{Retr}_x = \mathrm{Exp}_x$, for example Corollary 8 in (Zhang & Sra, 2016), we may get more elegant results by using $\mathrm{Exp}_x$ as retraction function in our algorithm.

The manifolds we encounter in optimization are mainly embedded sub-manifold and quotient manifold (Absil et al., 2009). Embedded sub-manifold is $F^{-1}(y)$ for a smooth function $F : \mathcal{M}_1 \to \mathcal{M}_2$, where $\mathcal{M}_1, \mathcal{M}_2$ are two manifolds and $y \in \mathcal{M}_2$. Quotient manifold is a quotient topology space generalized by a specific equivalence relationship $\sim$. In this paper, we use Oblique manifold and Grassmann manifold which are embedded sub-manifold and quotient manifold respectively.

The difficulty we faced in optimization on manifold is calculating tangent space $T_x\mathcal{M}$ and Riemannian gradient $\mathrm{grad} f(x)$. Giving a exact formula of a tangent space $T_x\mathcal{M}$ is not a easy problem. On the other hand, since Riemannian gradient is $\nabla f(x)$ projected to a tangent space $T_x\mathcal{M}$, finding projection matrix to a specific space $T_x\mathcal{M}$ is nontrivial.

### ACKNOWLEDGMENTS

Use unnumbered third level headings for the acknowledgments. All acknowledgments, including those to funding agencies, go at the end of the paper.

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

# A   DISCUSSION ABOUT THE FRAMEWORK OF GRADIENT DESCENT WITH DRIFT

## A.1   CONNECTION WITH CLASSICAL ALGORITHMS

In this section, we study the frame work of gradient descent with drift. In a special case, we regard $\mathbb{R}^n$ as a manifold. Then, Rienmann gradient $\mathrm{grad} f(x) = \nabla f(x)$, tangent space $T_x\mathcal{M} = \mathbb{R}^n$ and $\mathrm{Retr}_x(\eta) = x + \eta$. In Algorithm (1), we set

$$h_{k+1} = -\frac{1}{L}(a_k \nabla f(x_k) + b_k h_k),$$

where

$$\delta < a_k < 2, \qquad b_k = \frac{2(1-a_k)\langle \nabla f(x_k), h_k\rangle}{\|h_k\|^2}.$$

Then we have

$$x_{k+1} = x_k - \frac{1}{L}(a_k \nabla f(x_k) + b_k h_k)$$
$$= x_k - \frac{1}{L}(a_k \nabla f(x_k) - \frac{b_k}{L}(a_{k-1}\nabla f(x_{k-1}) + b_{k-1}h_{k-1})),$$

which is exactly a kind of gradient descent with momentum. And this algorithm is convergence as we proved. On the other hand, if choosing $h_k$ as gradient of a regularization term $R(x)$ on $x_k$. For example, $h_k$ becomes $2x_k$ when $R(x) = \|x\|^2$. The iteration point in Algorithm (1) is achieved by gradient descent with regularization term.

## A.2   A STOCHASTIC DRIFT

The drift in (equation 3) we have discussed is non-stochastic. But actually, we can change the drift as a stochastic term to construct a non-descent algorithm. Meanwhile, stochastic drift gives iteration sequence ability of jumping from local minimizer. The update rule is

$$x_{k+1} = \mathrm{Retr}_{x_k}[-\frac{1}{L}(a_k \mathrm{grad} f(x_k) + b_k P_{x_k}\xi_k),] \tag{10}$$

where $\xi_k$ is a random vector with mean vector $\mu$, covariance matrix $\Sigma$. The process of this algorithm is Algorithm 2.

---

**Algorithm 2** Non-descent method with stochastic noise

---

Input $0 < \delta < 2$, $x_0 \in \mathcal{M}$, $\mathrm{Retr}_x$, $\varepsilon > 0$
$k \to 1$
**while** $\|\mathrm{grad} f(x)\| > \varepsilon$ **do**
   Sample $\xi_k$ with mean vector $\mu$ and covariance matrix $\Sigma$
   $b_k = \frac{1}{k^2}$
   $a_k = \begin{cases} 1 - \frac{1}{k\|\mathrm{grad} f(x_k)\|} & \|\mathrm{grad} f(x_k)\| \geq \frac{1}{k} \\ \delta & \|\mathrm{grad} f(x_k)\| < \frac{1}{k} \end{cases}$
   $x_{k+1} = \mathrm{Retr}_{x_k}[-\frac{1}{L}(a_k \mathrm{grad} f(x_k) + b_k P_{x_k}\xi_k)]$        update step
   $P_{x_k}$ is projection matrix to tangent space $T_x\mathcal{M}$
   $k \leftarrow k + 1$
**end while**
**return** $x_k$

---

We give convergence theorem of Algorithm 2. The proof implies that this algorithm is non-descent, it also shows how we set $a_k$ and $b_k$.

**Theorem A.1** *For function $f(x) \geq f^* > -\infty$, and Lipschtiz gradient. If we construct the sequence $\{x_k\}$ like (equation 10), choosing $\{a_k\}$, $\{b_k\}$ satisfy $\sum\limits_{k=1}^{\infty} b_k < \infty$ and*

$$a_k = \begin{cases} 1 - \frac{1}{k\|\mathrm{grad} f(x_k)\|} & \|\mathrm{grad} f(x_k)\| \geq \frac{1}{k} \\ \delta & \|\mathrm{grad} f(x_k)\| < \frac{1}{k} \end{cases} \tag{11}$$

*where $0 < \delta < 2$, we have* $\liminf_{k\to\infty} \|\mathrm{grad} f(x_k)\|^2 = 0.$

In this theorem, $b_k$ control the speed of back fire. The noise $\xi_k$ in (equation 10) has small effect to iteration process when $k$ is large, because sequence is about to be stable after enough iterations. But in beginning of iteration procedure, noise $\xi_k$ effects much which give iteration sequence ability of jumping from local minimizer.

## B    PROOF OF THEOREMS

In this section, we give proof of theorems in this paper. The proof of Theorem 2.1 is

**Proof 4 (*proof of Theorem 2.1*)**  According to definition 2.2 of descent condition, we have

$$f(x_0) - f(x_k) \geq \sum_{i=0}^{k-1} a_i \|\mathrm{grad} f(x_i)\|^2,$$

for any $k$. Since $f$ is lower finite, we have

$$\sum_{i=0}^{\infty} a_i \|\mathrm{grad} f(x_i)\|^2 \leq f(x_0) - f^* < \infty,$$

where $f(x) \geq f^* > -\infty$, it means $\liminf_{k\to\infty} a_k \|\mathrm{grad} f(x_k)\| = 0.$    ∎

The proof of Theorem 2.2 is

**Proof 5 (*proof of Theorem 2.2*)**  Since $f$ satisfy Lischtiz gradient(2.1), we have

$$f(x_k) - f(x_{k+1}) \geq \frac{1}{L}(a_k - \frac{a_k^2}{2})\|\mathrm{grad} f(x_k)\|^2 + \frac{1}{L}b_k(1 - a_k)\langle \mathrm{grad} f(x_k), h_k\rangle$$
$$- \frac{b_k^2}{2L}\|h_k\|^2 .$$

By the definition of $b_k$, we got

$$f(x_k) - f(x_{k+1}) \geq (a_k - \frac{a_k^2}{2})\|\mathrm{grad} f(x_k)\|^2. \tag{12}$$

Since $a_k - \frac{a_k^2}{2} > 0$, by Theorem 2.1, we have

$$\lim_{k\to\infty} \inf (a_k - \frac{a_k^2}{2})\|\mathrm{grad} f(x_k)\|^2 = 0.$$

The definition of $a_k$ implies that $\|\mathrm{grad} f(x)\|^2 \to 0.$    ∎

The proof of Theorem 2.3 gives convergence rate of gradient descent on manifold with drift.

**Proof 6 (*proof of Theorem 2.3*)**  For any $\varepsilon > 0$, assuming $\|\mathrm{grad} f(x_i)\| \geq \varepsilon, i \in \mathbb{N}$ until $i = k$. Then

$$f(x_0) - f(x_k) \geq \sum_{i=0}^{k-1}[(\frac{1}{i+1} - \frac{1}{2(i+1)^2}) \vee (\delta - \frac{\delta^2}{2})]\|\mathrm{grad} f(x_i)\|^2$$

$$\geq [(\log{(k+1)} - \sum_{i=0}^{\infty} \frac{1}{2(i+1)^2}) \vee k(\delta - \frac{\delta^2}{2})]\|\mathrm{grad} f(x_i)\|^2,$$

here we use the relationship $\log{(1+x)} \leq x$ when $x \geq 0$. Since $\sum_{i=0}^{\infty} \frac{1}{2(i+1)^2} = \frac{\pi^2}{12} < \infty$(Zorich, 2002), for $i \leq k - 1$, we have

$$\log{(k+1)} \leq \frac{f(x_0) - f^*}{\|\mathrm{grad} f(x_i)\|^2} + \frac{\pi^2}{12}$$

and

$$k \leq \frac{f(x_0) - f^*}{(\delta - \frac{\delta^2}{2})\|\mathrm{grad} f(x_i)\|^2}.$$

Since $\|\mathrm{grad} f(x_i)\| \geq \varepsilon$ when $i \leq k - 1$, $k \leq (\lceil\exp{\{\frac{f(x_0)-f^*}{\varepsilon^2} + \frac{\pi^2}{12}\}}\rceil - 1) \wedge \lceil\frac{2(f(x_0)-f^*)}{\varepsilon^2(2\delta-\delta^2)}\rceil.$    ∎

Before proof Theorem A.1, we need two lemmas.

**Lemma B.1** *A random vector with $\mathbb{E}x = \mu$ and $\mathrm{Cov}x = \Sigma$. Then for any symmetric matrix A, we have*

$$\mathbb{E}(x^T A x) = \mu^T A \mu + \mathrm{tr}(A\Sigma).$$

This lemma can be derived from Wang et al. (2003)'s theorem 3.2.1 of Page 57. The other lemma is

**Lemma B.2** *If A and $\Sigma$ are $n \times n$ symmetric matrix, then $\mathrm{tr}(A\Sigma) \leq \mathrm{tr}(A)\lambda_{max}$, where $\lambda_{max}$ is the largest engine value of $\Sigma$*

**Proof 7 (*proof of Lemma B.2*)** According to spectral decomposition of symmetric matrix $A$, $A$ can be written as $\sum_{i=1}^{n} \lambda_i \gamma_i \gamma_i^T$, where $\lambda_i$ is engine value of $A$ and $\gamma_i$ is engine vector corresponding to $\lambda_i$. They satisfy

$$\gamma_i^T \gamma_j = \delta_{ij},$$

$\delta_{ij}$ is Kronecker delta. Hence

$$\mathrm{tr}(A\Sigma) = \sum_{i=1}^{n} \mathrm{tr}(\lambda_i \gamma_i \gamma_i^T \Sigma) = \sum_{i=1}^{n} \mathrm{tr}(\lambda_i \gamma_i^T \Sigma \gamma_i) \leq \lambda_{max} \sum_{i=1}^{n} \mathrm{tr}(\lambda_i) = \lambda_{max} \mathrm{tr}(A).$$

Here we use Rayleigh theorem of theorem 2.4.(Wang et al., 2003)

**Proof 8 (*proof of Theorem A.1*)** Since $P_{x_k}$ is a projection matrix, which is a symmetric idempotent matrix. Because $f$ satisfies Lipschtiz gradient(2.1), we have

$$f(x_1) - f(x_k) \geq \frac{1}{L} \left[ \sum_{i=1}^{k-1} (a_i - \frac{a_i^2}{2}) \|\mathrm{grad}f(x_i)\|^2 + b_i(1-a_i)\varepsilon_i - \frac{b_i^2}{2}\eta_i \right],$$

where $\varepsilon_i = (P_{x_i}\mathrm{grad}f(x_i))^T \xi_i = \mathrm{grad}f(x_i))^T \xi_i$, $\eta_i = \xi_i^T P_{x_i}^T P_{xi} \xi_i = \xi_i^T P_{x_i} \xi_i$. Due to the two random variables, algorithm (2) is not necessary descent. $\Sigma$ is a symmetric positive definite matrix. By Schwarz equality and definition of $a_i$, we have

$$\mathbb{E} \sum_{i=1}^{k-1} [b_i(1-a_i)\varepsilon_i] \leq \mathbb{E} \sum_{i=1}^{k-1} b_i(1-a_i)\|\mathrm{grad}f(x_i)\|\|\xi_i\|$$

$$\leq \sum_{i=1}^{k-1} b_i \left( \mathbb{E}(1-a_i)^2 \|\mathrm{grad}f(x_i)\|^2 \right)^{\frac{1}{2}} \left( \mathbb{E}\|\xi_i\|^2 \right)^{\frac{1}{2}}$$

$$= \sum_{i=1}^{k-1} b_i \left( \mathbb{E}(1-a_i)^2 \|\mathrm{grad}f(x_i)\|^2 \right)^{\frac{1}{2}} [\|\mu\|^2 + \mathrm{tr}(\Sigma)]^{\frac{1}{2}}$$

$$\leq [\|\mu\|^2 + \leq \mathrm{tr}(\Sigma)]^{\frac{1}{2}} \sum_{i=1}^{k-1} \frac{b_i}{i}.$$

By Fatou's lemma, we have

$$\mathbb{E} \liminf_{k \to \infty} [\sum_{i=1}^{k-1} b_i(1-a_i)\varepsilon_i] \leq \liminf_{k \to \infty} \mathbb{E}[\sum_{i=1}^{k-1} b_i(1-a_i)\varepsilon_i]$$

$$\leq \lim_{k \to \infty} [\|\mu\|^2 + \leq \mathrm{tr}(\Sigma)]^{\frac{1}{2}} \sum_{i=1}^{k-1} \frac{b_i}{i} = [\|\mu\|^2 + \leq \mathrm{tr}(\Sigma)]^{\frac{1}{2}} \sum_{i=1}^{\infty} \frac{b_i}{i} < \infty,$$

which implies

$$\liminf_{k \to \infty} \sum_{i=1}^{k-1} b_i(1-a_i)\varepsilon_i < \infty \qquad \text{a.s.}$$

Since $\text{tr}(P_{x_i}) = \text{rank}(P_{x_i})$ and the largest engine value of $P_{x_k}$ is 1, we have

$$\mathbb{E}(\eta_i) = \mathbb{E}(\xi_i^T P_{x_i} \xi_i) = \mu^T P_{x_i} \mu + \text{tr}(P_{x_i} \Sigma) \leq \|\mu\|^2 + \lambda_{max} d,$$

where $d$ is the dimension of $x$ and $\lambda_{max}$ is the largest engine value of $\Sigma$. By Levy's theorem, we have

$$\mathbb{E}[\lim_{k \to \infty} \sum_{i=1}^{k-1} \frac{b_i^2}{2} \eta_i] = \lim_{k \to \infty} \mathbb{E}[\sum_{i=1}^{k-1} \frac{b_i^2}{2} \eta_i] \leq \lim_{k \to \infty} \sum_{i=1}^{k-1} (\|\mu\|^2 + \lambda_{max} d) \frac{b_i^2}{2} < \infty,$$

which implies $\sum_{i=1}^{\infty} \frac{b_i^2}{2} \eta_i$ is finite almost surely. Hence

$$\sum_{i=1}^{\infty} (a_i - \frac{a_i^2}{2}) \|\text{grad} f(i_k)\|^2 \leq L(f(x_1) - f^*) - \liminf_{k \to \infty} \sum_{i=1}^{k-1} b_i(1 - a_i)\varepsilon_i + \sum_{i=1}^{\infty} \frac{b_i^2}{2} \eta_i < \infty \qquad \text{a.s.}$$

then we have

$$\liminf_{k \to \infty} (a_k - \frac{a_k^2}{2}) \|\text{grad} f(x_k)\|^2 = 0 \qquad \text{a.s.} \tag{13}$$

by the definition of $a_k$(equation 11), equation (13) is equivalent to $\liminf_{k \to \infty} \|\text{grad} f(x_k)\|^2 = 0$ almost surely. ∎

