# OpenReview forum: "Optimization on Multiple Manifolds"
_ICLR.cc/2019/Conference_

### Official Review · AnonReviewer2 · 2018-10-24
**The problem considered in this paper is interesting. However, there are quite a few fundamental errors about Riemannian optimization. Therefore, I donot think this paper can be published at this stage.**

**Rating:** 3
**Confidence:** 4

**Review:**

This paper considers an optimization problem defined on the intersection of multiple manifolds and the intersection is not a manifold. An optimization algorithm is proposed and its convergence analysis is given. An experiment of neural network with batch normalizatiion is used to demonstrate the performance of the algorithm.

The problem considered in this paper is interesting. However, there are quite a few fundamental errors about Riemannian optimization. In addition, the convergence analysis is not complete. See more details below. Therefore, I donot think this paper can be published at this stage.

*) P2, Section 2.1, line 2: The statement "A manifold is a subspace of R^n$ is not true in general.
*) P2, Section 2.1, line 7: The statement "manifold is not a linear space" is not true in general. A manifold can be a linear space, such as the vector space R^n.
*) P2, Section 2.1, below (2): The statement "Riemannian gradient is the orthogonal projection of gradient \nabla f(x) ..." is not true in general.
*) P3, (3) and (4): what is the definition of $h_k^{(1)}$ and $h_k^{(2)}$. Are they arbitrary or the ones given on Page 4?
*) P4, Theorem 2.3: the iterates {x_k} converges in the sense that \|gradf(x_k)\| goes to 0. Does {x_k} go to the intersection of the two manifolds \mathcal{M}_1 and \mathcal{M}_2? To complete the proofs, the author may need to show that \|gradf(y_k)\| goes to 0 and {x_k} and {y_k} have the same limit.
*) P5, the grassmann manifold with p = 1: G(1, n), is called projective space, and the Stiefel manifold with p = 1: St(n, 1) is called the unit sphere.
*) P6, the discussion of the intersection of G(1, n) and St(n, 1) does not make sense to me. G(1, n) is a quotient manifold, which is not a submanifold of R^n. Given a quotient manifold, the typical way in optimization framework is to choose representation of the quotient manifold. Fortunately, the projective space has a global orthogonal section, which is the unit sphere. In other words, G(1, n) is diffemorphisic to the unit sphere St(n, 1), and even can be isometric if appropriate Riemannian metrics are used on G(1, n) and St(n, 1). Therefore, I don't understand the notion of the intersection of G(1, n) and St(n, 1).

---

### Official Review · AnonReviewer1 · 2018-11-01
**The problems addressed in the paper are interesting and crucial from both practical and theoretical perspectives. However, there are various major mathematical, conceptual and algorithmic problems with this paper.**

**Rating:** 1
**Confidence:** 5

**Review:**


The title is misleading, since only two particular manifolds are studied in this work. In addition, the proposed methods cannot be applied to a larger or a general class of manifolds. Therefore, you should update the title.

There are multiple problem definitions proposed in the paper. They are not compatible with each other and also with the proposed methods. In addition, some of the proposed problem definitions are incorrect, as explained below:

You should be more precise about the definition of the manifold you consider in this paper. For example, in equation (1), please define your manifold of interest more precisely checking some standard textbooks.

Please define intersection of manifolds, what do you mean by which intersection of which type of manifolds?

In the contribution (1); the paper does not introduce an algorithm to deal with optimization on with multiple manifolds, but for a particular type of individual manifolds.

In the contribution (2): It is not clear why and how the proposed method can be applied to optimization on manifolds with momentum (what do you mean by use of momentum here?), and regularization (what do you mean by regularization?). There are many problems with this claim, but you can simply consider that applying momentum and regularization will affect the geometry of loss landscape.

Definition of retraction is not precise, please fix it.
What is L in equation (2)?

Please define neighborhood in U_x in Lemma 2.1.

What is || ||in Lemma 2.1?

As you also noticed on page 3, an intersection of manifolds may not be a manifold. Then, your proposed first problem (1) fails. Therefore, you should completely change your claims on your problem definitions and contributions.

What do you mean by “We add a drift which contains information from the other manifold to the original gradient descent on manifold”? What is “the information from the manifold”? In equations (3) and (4), you just apply optimization on manifolds individually.

How do you compute/determine a_k^(1) and a_k^(2)? How do they affect the theoretical and experimental results?
In your claim “From the construction of bk, we can see that the smaller the correlation between gradf(xk) and hk is, the smaller effect the information from M2 brings”, it is not clear how “the information from M2” affects? First, again, what is “the information”? Second, b_k^(1) and b_k^(2) are computed for individual manifolds separately. Then, how “the information” make an effect?

 In Theorem 2.2, what do you mean by “then xk convergence to a local minimizer“?

What is <,> in Theorem 2.2?

What is ^ in Theorem 2.3?

What is v in proof 6?

What is an engine value?

What does P (1) xk gradf(yk) denote in computation of h_k? For example, gradf(yk) is a vector on tangent space of the second manifold at yk. Then, how do you project orthogonally this projected vector to the tangent space of the first manifold at xk?

They may be completely different geometries, and such an “orthogonal projection” may not exist in general. Then, how do you compute and calculate that projection?

All the theoretical results given in the paper are not about convergence of parameters on a manifold at the intersection or product of manifolds but for an individual manifold. For example, x and y belong to manifolds M1 and M2, and convergence results is about x. How are they related to parameters at the intersection or product of manifolds?

The statements regarding batch normalization are confusing and also sound incorrect:

Do you apply batch normalization on weights on BN(w)?

Please explain what you mean by “BN(w) has same image space on G(1, n) and St(n, 1)“. There are not such results in the papers Cho & Lee (2017); Huang et al. (2017) you cited for these results.

What do you mean by “applying optimization on manifold to batch normalization problem”?

In your statement “However, the property of these two manifold implies that we can actually optimize on the intersection of two manifolds”. Please explain how does this property imply this result more precisely?

Please define “Grassmann manifold G(1, n)“ more precisely. In your notation, together with explanation of the notation for St(n,p), G(1,n) is like a set of 1xn dimensional row vectors, while St(n,1) is an nx1 dimensional column vector, Then, their intersection is an empty set and your proposal for optimization on a vector on their intersection is wrong.

Notation and definitions used in (9) are wrong and confusing. Please check and revise them.

In the whole paper, the problem, method, solutions, theorems, and contributions are proposed for optimization using parameters which belong to intersection of some manifolds. Then, suddenly, you start considering optimization on product manifolds, and give the results for that;

What does the statement “Then we apply Algorithm 1 to update parameters, which means we optimize on a product manifold” mean?

What do “G(1, k1) × · · · G(1, kn)” and “St(k1, 1) × · · · St(kn, 1)” denote?

Don’t you perform optimization on intersection of manifolds? Why do you ignore your original problem and methods, and consider this problem?

In addition, how do you use your Algorithm 1 for optimization on product manifolds? Optimization on intersection on manifolds and product manifolds are completely different problems. If they are same or related to each in particular cases in your specific definitions, then you should provide these definitions more precisely.

What do you mean by optimization on product manifold of weights of all layers? If you compute a product manifold for spaces of all layers, then you simply perform a shallow optimization on a huge matrix containing millions of dimensions according to this definition. First, how do you do that? Second, how can you train a large network using this approach?

In the experiments, please first give variance of errors. These results are statistically insignificant.

Which problem is solved to perform these experiments is not also clear (see above).

The results reported in the paper are also not good, may be due to the mathematical and algorithmic  problems and errors mentioned above. Please clarify them, and provide additional results, especially using other datasets (small scale mnist and large scale imagenet), and networks (mlp, vgg, resnet etc.)

Related work is also incomplete, such that many traditional and recent work on optimization on multiple manifolds are omitted.

---

### Official Review · AnonReviewer3 · 2018-11-05
**a novel algorithm on optimization on multiple manifolds**

**Rating:** 7
**Confidence:** 3

**Review:**

The paper proposes a novel algorithm for optimization on multiple manifolds. The moving direction fuses gradient information from each manifolds via correlation. More importantly, the convergence is guaranteed.

However, the empirical results seems not very good compared to SGD.

My concerns:

1) How you ensure each step is descent?

2) How is the performance of the proposed algorithm compared to the ADMM which is well-suited for this problem.

The presentation needs to be improved.

---

### Public Comment · (anonymous) · 2018-10-11
**gradient descent on manifold**


The optimization with many constraints is usually a very difficult problem.  I don't fully understand details in your paper, still have a question on the iterate relation in the paper that x_{k+1} = Retr_x(-1/L(grad(f(x_k)))).  Say if the constraints are all equalities, then these constraints will confined the feasible domain to many hyperplanes. And then when we calculate the Riemannian gradient of f(x), how to make sure that the update point x_{k+1} locates inside the feasible domain? Or say how to guarantee that the update point x_{k+1} always move on manifolds?

---

> ### Public Comment · (anonymous) · 2018-10-15
> **reply to gradient descent on manifold**
>
> Thanks for your attention to this paper. For the manifold optimization with a single manifold, to ensure that the iterative point x_{k+1} always moves on manifold, we need the operator $Retr_{x}$: $Retr_{x}(v)$ is a map from $T_{x}\mathcal{M}$ to manifold $\mathcal{M}$, where $T_{x}\mathcal{M}$ is the tangent space at point $x\in\mathcal{M}$. Since ${\rm grad}f(x)$ is on $T_{x}\mathcal{M}$, $x_{k+1} = Retr_x_{k}(-1/L({\rm grad}f(x_k))$ must live on manifold.
> In fact, the iterative points obtained by the updating rule $x_{k+1} = Retr_{x_{k}}(v_{k})$ locate on manifold as long as $v_{k}\in T_{x_{k}}\mathcal{M}$. Since the updating rule in our paper satisfy $v_{k}\in T_{x_{k}}\mathcal{M}$, iterates must live on the corresponding manifold.
> For the case with multiple manifold where the optimization region corresponds to the intersection of these manifolds ,  existing methods cannot guarantee that iterates live on the intersection of manifolds, because intersection of manifolds may not be a manifold. They can only reach minima on each manifold rather than the intersection of them. In contrast, our method heuristically sets the updating direction on one manifold as the manifold gradient perturbed by the gradient information from other manifolds. This updating rule renders the common minima which is shared by all the manifolds, if there exists, as the only stable points of our algorithm. This is a heuristic argument rather than a theoretical analysis.

---

### Meta-Review · Area_Chair1 · 2018-12-17
**Paper requires further refinement**

**Confidence:** 5
**Recommendation:** Reject

**Metareview:**

The paper describes a constrained optimization strategy for optimizing on an intersection of two manifolds.  Unfortunately, the paper suffers from generally weak presentation quality, with the technical exposition seriously criticized by two out of the three reviewers.  (The single positive review is too short and devoid of content to be taken seriously.  Even there, concerns are expressed.) This paper requires substantial improvement before it could be considered for publication.